# Is Anyone There? Learning a Planner Contingent on Perceptual Uncertainty

**Charles Packer**
UC Berkeley

**Nicholas Rhinehart**
UC Berkeley

**Rowan McAllister**
Toyota Research Institute

**Matthew A. Wright**
UC Berkeley

**Xin Wang**
Microsoft Research

**Jeff He**
UC Berkeley

**Sergey Levine**
UC Berkeley

**Joseph E. Gonzalez**
UC Berkeley

**Abstract:** Robots in complex multi-agent environments should reason about the intentions of observed and *currently unobserved* agents. In this paper, we present a new learning-based method for prediction and planning in complex multi-agent environments where the states of the other agents are partially-observed. Our approach, Active Visual Planning (AVP), uses high-dimensional observations to learn a flow-based generative model of multi-agent joint trajectories, including unobserved agents that may be revealed in the near future, depending on the robot's actions. Our predictive model is implemented using deep neural networks that map raw observations to future detection and pose trajectories and is learned entirely offline using a dataset of recorded observations (not ground-truth states). Once learned, our predictive model can be used for contingency planning over the potential existence, intentions, and positions of unobserved agents. We demonstrate the effectiveness of AVP on a set of autonomous driving environments inspired by real-world scenarios that require reasoning about the existence of other unobserved agents for safe and efficient driving. In these environments, AVP achieves expert-level closed-loop performance, while methods that do not reason about potential unobserved agents exhibit either overconfident or underconfident behavior.

**Keywords:** Forecasting, Planning, Partial Observability, Autonomous Driving

## 1 Introduction

When navigating in complex multi-agent scenarios, humans reason about the uncertainty of their own perception *and* how their actions affect their perception [1]. For example, a human driver at a blind intersection depicted in Fig. 1a may inch forward to improve their visibility of oncoming traffic before deciding whether or not to proceed. This type of explicit reasoning is a natural human behavior critical to safe and efficient decision making. However, it does not exist in autonomous systems that assume full observability of the environment, nor in systems that assume their limited situational awareness is persistent and unactionable. While it is possible to engineer human-like behavior for many individual scenarios involving perceptual uncertainty, can we instead develop a principled data-driven solution to learn end-to-end planners that incorporates perceptual uncertainty?

The partially observable Markov decision process (POMDP) model admits a notion of perceptual uncertainty, where the state of the underlying MDP is hidden due to imperfect perception (such as visual occlusions or sensor noise) and must be estimated by the agent in order to construct an optimal policy. POMDPs can be solved exactly for small discrete state-action spaces, but currently cannot be in large or continuous state-action spaces typical of AV planning [13], leading to a variety of approximate solutions that require significant human design effort, e.g. state spaces, affordances, scenario-specific heuristics, and/or dynamics models. Reinforcement learning (RL) can be applied to POMDP problems without requiring human-defined states or dynamics, however, RL adds a significant burden of online data collection. We propose a tractable data-driven method to accommodate partial observability, unknown dynamics, raw observations, and offline data.

6th Conference on Robot Learning (CoRL 2022), Auckland, New Zealand.

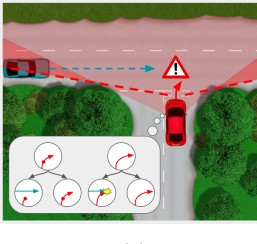

| Planning method | Learned model | Learned controller | Raw obs. as input | Partial observ. | Contingency in observ. | Closed-loop env. evaluation |
|---|---|---|---|---|---|---|
| Fisac et al. [2] | ✗ | None | ✗ | ✗ | None | ✓ |
| Bajcsy et al. [3] | ✓ | Offline | ✗ | ✗ | None | ✓ |
| Ivanovic et al. [4] | ✓ | Offline | ✗ | ✗ | Active | ✗ |
| Cui et al. [5] | ✓ | Offline | ✓ | ✗ | Active | ✗ |
| Rhinehart et al. [6] | ✓ | Offline | ✓ | ✗ | Active | ✓ |
| Tas et al. [7] | ✗ | None | ✗ | ✓ | None | ✓ |
| McGill et al. [8] | ✗ | None | ✗ | ✓ | None | ✓ |
| Wang et al. [9] | ✗ | None | ✗ | ✓ | Active | ✓ |
| Andersen et al. [10] | ✗ | None | ✗ | ✓ | Active | ✓ |
| Zhang et al. [11] | ✗ | None | ✗ | ✓ | Active | ✓ |
| Wray et al. [12] | ✗ | None | ✗ | ✓ | Active | ✓ |
| AVP (*ours*) | ✓ | Offline | ✓ | ✓ | Active | ✓ |

| (a) | (b) |
|---|---|

**Figure 1:** **(a)** Navigating a blind intersection requires prediction and planning under perception-induced uncertainty: an expert driver (red) should not only predict that a vehicle (blue) may emerge from the occluded area, but also understand that inching forward into the intersection will reduce uncertainty regarding oncoming traffic by improving visibility, thus enabling the driver to safely execute a turn. *Active Visual Planning* (*AVP*) enables prediction and planning with respect to unobserved agents, including actively planning to reduce future uncertainty due to perception. **(b)** Comparison of recent planning approaches for autonomous vehicles, focused on data-driven contingency planning (upper section) and planning under perceptual uncertainty (lower section).

In this paper, we present Active Visual Planning (AVP), a scalable method for contingency planning under multiple forms of partial observability (uncertainty in existence, positions, and intentions) that accommodates unknown dynamics, raw high-dimensional observations such as camera images or LiDAR scans, and learning from offline data. AVP models discrete uncertainty (uncertainty in existence) and continuous uncertainty (uncertainty in position) by reasoning about the existence of other entities and their future trajectories and planning accordingly. Importantly, AVP incentivizes an agent to take actions that actively reduce uncertainty over the existence of unobserved entities (e.g., moving to a location with a wider field of view) if it improves the quality of the overall plan (only a small subset of all uncertainty induced by imperfect perception is relevant to near-term decision making in many real-world scenarios). We evaluate AVP in various multi-agent settings in the CARLA [14] simulation environment. Our main contributions are:

- The first fully-learned planner that supports active contingency planning over perceptual uncertainty.
- A benchmark for evaluating the robustness of planners to real-world examples of perceptual uncertainty in a visually realistic simulator (CARLA). We illustrate that our method outperforms methods that do not properly account for potential occluded vehicles. Supplementary material can be found at https://sites.google.com/view/active-visual-planning.

## 2 Related Work

**POMDPs and belief space planning.** A long history of research in POMDP solvers and belief space planning for autonomous driving exists [15, 16, 17], and recent work has demonstrated success in modern autonomous vehicle (AV) simulators and real-world systems [18, 19, 20, 21, 22, 23]. Simple POMDPs can be solved using brute force enumeration [24], incremental pruning [25], and linear support [26], but in large or continuous state-action spaces typical of AV planning, exact POMDP solutions are generally intractable [13], leading to approximations such as Point Based Value Iteration [27, 28] and Heuristic Search Value Iteration [29], or making simplifying assumptions for tractability such as assuming belief distributions are unimodal, as iterative-linear (e.g. iLQG controllers) or moment-matching methods [30] do. Sampling-based Monte Carlo tree search methods [31, 32, 33] improve on point-based solvers and enable scaling to larger POMDPs with continuous state and action spaces. However, approximate methods that have been shown to work on real AV systems require human-defined state spaces, affordances, and scenario-specific heuristics [12] (e.g. augmenting states with affordances such as time since last stop, distance to stop sign, etc). In contrast, our modeling framework (AVP) is data-driven and automatically learns both a dynamics model and contextual features from raw sensor observations, and we demonstrate that the same model architecture and planner can generalize to several different scenarios without modification. We illustrate these differences in Fig. 1b.

**Reinforcement and imitation learning.** Although the MDP framework of reinforcement learning (RL) assumes the agent receives state observations, POMDPs can be reduced to MDPs by recording

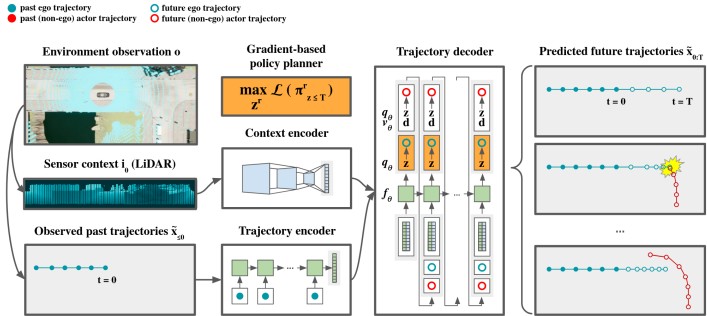

**Figure 2:** AVP uses the perceptual context and robot's past positions to predict both *when* agents may be detected and *where* the detected agents and the robot may go. By modeling both the motion of other vehicles and potentially detecting them as dependent on the robot's future position, the model enables the robot to factor in partial-observability when planning. The perceptual context $\mathbf{i}_0$ (e.g., LiDAR) is encoded by a CNN and the past positions, $\tilde{\mathbf{x}}_{\leq 0}$, are encoded by a GRU. Both encodings are input to a GRU decoder which generates a predicted distribution of future states $\mathbf{X}_{1:T}^{0:A}$ for each agent and detections $\mathbf{D}_{1:T}^{1:A}$ for each non-ego agent.

the action-observation history (or a sufficient statistic of the history) [34], and in practice deep RL has been shown to work well in certain POMDP environments using truncated histories and recurrent neural networks [35]. Both model-free [36, 37, 38, 39] and model-based deep RL [40] have been successfully applied to AV tasks to plan directly from images. While RL requires learning *online*, our approach is trained fully offline from logged data. Recent work in offline RL [41] aims to use RL on previously collected data, but both offline and online RL require access to a reward function during training, whereas AVP does not. AVP is closely related to imitation learning (IL), which also learns a control policy offline [42, 43]. However, unlike IL, AVP only requires expert *observations* instead of expert *demonstrations* (i.e. AVP does not require action labels) and is flexible to planning new goals at test-time, as opposed to IL which directly learns a control-level policy.

**Occlusion-aware prediction and planning.** In settings with visual occlusions, map knowledge and geometry can be employed for occlusion-aware prediction [44, 45, 46] and planning [47, 48, 7, 8, 9, 10]. For example, forward reachable set (FRS) methods avoid the entire set of future reachable states of hypothetical vehicles placed on the edges of occluded regions. However, such approaches often result in overly conservative plans, not only by assuming the worst case in existence and position, but also by assuming the worst case in intention (i.e., that entities are controlled by adversarial policies). In contrast, AVP learns a probabilistic model of existence, positions, and intention, enabling our planner to weigh risk and reward instead of relying on worst-case analysis. AVP also does not utilize any explicit map knowledge or geometry (such as estimated field of view), and instead learns to use raw sensor observations. Game theory has been applied to occlusion-aware planning to improve on worst-case analysis by modeling interactions between vehicles [11]. Although game-theoretic planners are capable of generating expert-level driving policies, in practice, they are difficult to apply to scenarios with more than two vehicles, whereas AVP naturally scales beyond two agents.

## 3 Active Visual Planning

**Preliminaries:** We consider multi-agent settings comprising an unknown number of agents, which we model as a fixed-size large set of agents, with some potentially not present. We will subsequently refer to the agent that is guaranteed to exist as the robot *ego*, for which we are designing a controller, and $A$ other agents as human *actors* (no direct control). We consider continuous positions in $d$-dimensional Cartesian space and discrete time, and denote the position of agent $a$ at time $t$ as $\mathbf{x}_t^a \in \mathbb{R}^d$, and omit the superscript to denote the joint positions of all agents as $\mathbf{x}_t \in \mathbb{R}^{A \times d}$. Let $t = 0$ denote the current timestep, $\mathbf{x}_0$ denote the current joint positions, and $\mathbf{x}_{\leq t}$ denote future joint positions from $1$ to $t$. We say non-ego actors $\mathbf{x}_t^{1:A}$ are observed if and only if the actor is detected by the ego. We denote if an actor $a$ has been detected by ego at time $t$ with a binary flag $\mathbf{d}_t^a$, and the one-hot vector of all detections as $\mathbf{d}_t \in \mathbb{R}^A$; $\mathbf{d}_t^a = 1$ if actor $a$ has been observed in the ego agent's perceptual context $\mathbf{i}$ (e.g., LiDAR sensors or camera images). $\tilde{\mathbf{x}}_t$ denotes an *observed* joint position from the ego's perspective, where for each agent $a \neq 0$, the true position is revealed ($\tilde{\mathbf{x}}_t^a = \mathbf{x}_t^a$) if $\mathbf{d}_t^a = 1$, and is

masked otherwise. The robot's current observation is perceptual context and a subset of the past joint trajectories, denoted $\mathbf{o} \doteq (\tilde{\mathbf{x}}_{\leq 0}, \mathbf{i}_0)$. Uppercase denotes random variables, lowercase denotes realized values: $\mathbf{X}_{\leq t}$ and $\mathbf{D}_{\leq t}$ represent a stochastic sequence of future positions and future detections.

**A fully-observed model (prior work):** We base our models on prior work that introduced an autoregressive probabilistic model for multi-agent prediction [49] and planning [6] with several desirable properties: the predictive model can be fully learned from observations using maximum likelihood estimation (no need to specify an approximate dynamics model), it supports multi-modal joint predictions over varying numbers of agents, models co-dependencies between different agents, and enables straightforward yet expressive planning through gradient-based optimization of the model. Importantly, the plans generated through the model are *contingent*, i.e., the plans explicitly account for (are contingent on) different future outcomes of modelled stochastic events. However, these modelled events did *not* include the potential appearance of new agents, which precludes explicitly reasoning about such events. In many real-world driving scenarios, full observability is an unrealistic assumption: occlusions frequently block the view of other relevant agents, and the imperfect sensing and perception systems used in modern self-driving cars can often lead to low confidence or erroneous object detection and tracking. Our model enables prediction and planning with respect to currently-unobserved agents.

The starting point for our behavioral model is a parameteric probabilistic model $q_\theta(\mathbf{X}|\mathbf{o})$ of future joint trajectories $\mathbf{X}_{\leq t}$, conditioned on $\mathbf{o}$ (recall $\mathbf{o}$ is perceptual context $\mathbf{i}_0$ and past trajectories $\mathbf{x}_{\leq 0}$). The distribution is factorized across both time $(1...T)$ and agents $(0...A)$ into a product of autoregressive normal distributions $q_\theta^a$ of present positions $t$ for each agent $a$, and $q_\theta^a$ is conditioned on a paramaterized transformation of previous predictions and the context, denoted as $\mathbf{s}_{t-1} \doteq f_\theta(\mathbf{x}_{<t}^{1:A}, \mathbf{d}_{<t}^{1:A}, \mathbf{i})$:

$$q_\theta^{\text{full-obs.}}(\mathbf{X}_{\leq T}^{1:A}|\mathbf{o}) = \prod_{t=1}^{T} \prod_{a=0}^{A} q_\theta^a(\mathbf{X}_t^a; \mathbf{s}_{t-1}), \quad (1) \qquad q_\theta^a = \mathcal{N}(\mathbf{X}_t^a; \mu_\theta(\mathbf{s}_{t-1}), \sigma_\theta^2(\mathbf{s}_{t-1})), \quad (2)$$

where $\mathbf{X}$ can be sampled by drawing $\mathbf{Z} \sim \mathcal{N}(0, I)$: $\mathbf{X}_t^a = \mu_\theta(\mathbf{s}_{t-1}) + \sigma_\theta(\mathbf{s}_{t-1}) \cdot \mathbf{Z}_t^a$. This particular parameterization allows us to express variation in position as a probabilistic noise component drawn from a simple base distribution $\mathbf{Z}$; for the ego vehicle, we assume this noise is controllable (and thus fix $\mathbf{z}^0$ while planning), whereas for non-ego actors we sample $\mathbf{z}^{1:A} \sim \mathbf{Z}^{1:A}$.

**Modeling partial observability:** To adapt the model in Eq. 1 to the partially observed setting where many entries of $\mathbf{x}$ are often unknown, we model joint trajectories $\tilde{\mathbf{x}}$ of detected agents indices $\mathcal{D} \doteq \{(t, a) : \mathbf{d}_t^a = 1\}$ as $q_\theta^{\text{partial-obs.}}$, and model *the joint probability of future agent detection* as $v_\theta$:

$$q_\theta^{\text{partial-obs.}}(\tilde{\mathbf{X}}_{\leq T}^{1:A}|\mathbf{o}) = \prod_{(t,a) \in \mathcal{D}} q_\theta^a(\tilde{\mathbf{X}}_t^a; \mathbf{s}_{t-1}), \quad (3) \qquad v_\theta(\mathbf{D}_{\leq T}^{1:A}|\mathbf{o}) = \prod_{t=1}^{T} \prod_{a=1}^{A} v_t^a(\mathbf{D}_t^a; \mathbf{s}_{t-1}), \quad (4)$$

where $v_t^a$ is a Bernoulli distribution predicted by a neural-network that represents the probability that agent $a$ is detected at time $t$. Note $\mathbf{d}$ is observed during training (even when $\mathbf{x}$ is not). Since we no longer assume access to $\mathbf{x}$, $\mathbf{s}_{t-1}$ is instead a transformation of $\tilde{\mathbf{x}}$. Because $v_t^a$ is conditioned on the perceptual context, the model can relate visual cues (such as visibly occluded areas of a LiDAR point cloud or camera image) with the probability of future detection.

**Benefits of modeling *detection* instead of *existence*:** Eq. (4) models currently undetected agents appearing into view, and also currently detected agents disappearing from view. For example, this behavior is seen in the overtake and intersection environments in Fig. 3: if the ego agent successfully navigates past the other agent, the non-ego agent will eventually be predicted to "disappear" (become undetected) as it drives out of the ego vehicle's LiDAR range. By modeling detection instead of existence, we can train a partially-observed trajectory model without requiring privileged access to ground truth position information for all occluded agents; a learned model of existence similar to the model of detection in Eq. (4) (via a Bernoulli variable) would require explicitly modeling the positions of agents while undetected. AVP implicitly considers the dynamics between detected and undetected agents and retains aleatoric uncertainty in each distribution as a function of the observed

---

**Algorithm 1:** Planning with AVP

---

**Require:** Trained predictive model $M$ (consisting of a context encoder and autoregressive state encoders and decoder), partially observed past joint trajectories $\tilde{\mathbf{x}}_{\leq 0}$, present perceptual context $\mathbf{i}_0$, and planning loss criterion $\mathcal{L}$

1   Randomly initialize sequence of robot $\mathbf{z}$'s (each $\mathbf{z}_r$ denoted as $\mathbf{z}_t^0$)
2   Compute initial state encoding $\mathbf{s}_0$ using learned trajectory and context encoders on $\{\tilde{\mathbf{x}}_{\leq 0}, \mathbf{i}_0\}$
3   **for** each optimization step **do**
4      **for** each sample **do**
5          **for** each $t$ in horizon $T$ **do**
6              Pass the state encoding $\mathbf{s}_{t-1}$ to the learned trajectory decoder to get $\mathbf{D}_t^{1:A}$, $\mathbf{X}_t^{1:A}$, and RNN hidden state $h_t$
7              Sample $\mathbf{z}_t^{1:A} \sim \mathcal{N}(0, I)$ for every non-controllable (i.e., human) actor to compute $\mathbf{x}_t^{1:A}$ (use fixed $\mathbf{z}_t^0$ to compute $\mathbf{x}_t^0$)
8              Sample $\mathbf{d}_t^{1:A} \sim \mathbf{D}_t^{1:A}$ and combine with $\mathbf{x}_t^{0:A}$ to compute the observed positions $\tilde{\mathbf{x}}_t^{0:A}$
9              Append $\mathbf{d}_t^{1:A}$ and $\tilde{\mathbf{x}}_t^{0:A}$ to $h_t$ to encode $\mathbf{s}_t$.
10      Compute the gradient of the joint trajectory loss $\nabla \mathcal{L}(\pi_{\mathbf{z}_{\leq T}^r}^r)$ using sampled joint trajectories $\tilde{\mathbf{x}}$
11      Use $\nabla \mathcal{L}$ to update robot's $\mathbf{z}^0$

---

information (perceptual context and positions of detected agents); see the Fig. 3 in the supplement for a visual explanation.

**Planning with a learned model:** To use our learned model for planning, we plan a sequence of $\mathbf{z}$'s for the controllable robot agent, while sampling $\mathbf{z}$'s for all other human-controlled agents. By planning in $\mathbf{z}$-space, we are effectively planning *policies*, or contingent plans: fixing $\mathbf{z}^r$ creates a deterministic robot policy from a partial policy $\pi^r \doteq q_\theta^0$, which is specified by the learned parameters $\theta$ of the model. Because $\mathbf{z}^h$ remains sampled, the overall system remains stochastic; planning over $\mathbf{z}^h$ erroneously implies that all agents are explicitly controllable. Following [6], planning is derived as maximum a posteriori estimation of the free policy parameters $\mathbf{z}_{\leq T}^r$ conditioned on the initial positions and perceptual context, as well as a set of high-level goals $\mathcal{G} = (\mathbf{g}, \mathbb{G})$ consisting of target coordinate locations $\mathbf{g}$ and constraints $\mathbb{G}$ on the joint trajectory. In our experiments we apply a minimum distance constraint (8 meters to the nearest vehicle), which corresponds to a near-collision penalty. Estimation of posterior $p(\mathbf{z}_{\leq T}^r | \mathcal{G}, \mathbf{o})$ is a relatively lightweight gradient-based optimization procedure, since we optimize a small number of free parameters $\mathbf{z}^r$ while keeping all other parameters of the learned model fixed. For our cost function $\mathcal{G}$, we adopt the same lower bound planning objective as [6], which balances (1) a behavioral prior, (2) a scaled distance to the destination, and (3) a set of indicator functions $\delta_{\mathbb{G}}(\mathbf{x})$ on the optional constraints:

$$\mathcal{L}(\pi_{\mathbf{z}_{\leq T}}^r) = \mathop{\mathbb{E}}_{\mathbf{z}^h \sim \mathcal{N}(0,I)} \Big[ \log \underbrace{q_\theta^{\text{partial-obs.}}(\bar{\mathbf{x}}_{\leq T} | \mathbf{o})}_{(1)\ \text{prior}} + \log \underbrace{\mathcal{N}(\bar{\mathbf{x}}_T^r; \mathbf{g}, I)}_{(2)\ \text{destination}} + \log \underbrace{\delta_{\mathbb{G}}(\bar{\mathbf{x}}_{\leq T})}_{(3)\ \text{constraint}} \Big] \qquad (5)$$

Optimizing $\mathbf{z}_{\leq T}^r$ according to Eq. 5 thus produces a policy that (with high expectation) reaches a destination specified at test-time, while mimicking human-like driving behavior seen in the expert observations. The optimization of Eq. 5 accounts for the detection model because it is a function of $\tilde{\mathbf{x}}$ (note that $\mathbf{o}$ decomposes to $\{\tilde{\mathbf{x}}, \mathbf{d}, \mathbf{i}\}$), which itself is a function of the agent detection model (see Eq. 3). See Algorithm 1 for pseudocode describing the planning procedure.

**Model architecture and training:** We implement autoregressive models $q_\theta$ and $v_\theta$ using an encoder-decoder architecture (see Fig. 2) by building on the open-source implementation of [6]. $f_\theta$ is implemented as two separate networks: a convolutional neural network (CNN) that encodes the perceptual context, and a recurrent neural network (RNN) trajectory encoder that reads a past trajectory and generates an encoding. $q_\theta$ is implemented as an RNN decoder that takes the encoding and generates a sequence of $\mu$ and $\sigma$ representing future positions. Similarly, $v_\theta$ is implemented as an RNN that generates a sequence of Bernoulli parameters $p$. The entire network is jointly trained on the dataset of observations (consisting of observed joint trajectories and perceptual context) using maximum likelihood estimation according to Eqs. (3) and (4). During inference, if the $v_\theta$ does not predict a detection ($\mathbf{d}^h \sim v_\theta = 0$), then the predicted position (sampled from $q_\theta$) is ignored. AVP can handle scenarios with varying numbers of agents as long as the model was trained on a dataset that includes the same number of agents, as shown by the three-agent intersection experiment. Beyond this practical example of AVP's efficacy, theoretically, an agent trained on a dataset containing up to N agents will be able to handle up to N agents at test-time, and the variability in number of agents can be handled through the modeling of detection (the majority of N agents will then be predicted as undetected). In our experiments, we use a CNN with three convolutional layers (each with 32 filters

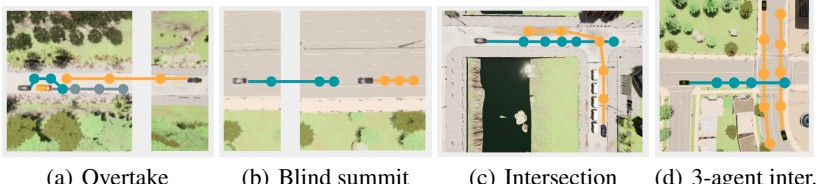

| (a) Overtake | (b) Blind summit | (c) Intersection | (d) 3-agent inter. |

**Figure 3:** An overhead visual of each scenario implemented in CARLA. An example joint trajectory is plotted to illustrate the expert human policy in the presence of an occluded actor. Trajectories for each car are visualized using circles and lines (ego in blue, actor in orange, and other moving entities in grey), with the circles evenly spaced in time. The overtake and blind summit scenarios have portions of the map cropped to fit the page.

and an $8 \times 8$ kernel) as our context encoders, and GRUs with three layers (256 hidden units each) as our recurrent encoders and decoders. The initial observation (model input) during model training contains a $60 \times 360$ LiDAR range map and a partially observed trajectory with 15 waypoints, sampled at 20Hz. The future trajectory (target sequence) consists of 30 waypoints, sampled at 7.5Hz. Further details can be found in the supplemental material.

## 4   Experiments

Scenarios requiring decision-making under perceptual uncertainty are ubiquitous in the domain of autonomous vehicles: occlusions obscuring important objects and agents are unavoidable even with state-of-the-art LiDAR, RADAR, and image-based sensors, additionally, sensor performance can significantly degrade under certain conditions such as inclement weather. To test the capability of our method, we focus on several autonomous driving scenarios where correctly reasoning about the existence of occluded agents leads to a clear difference in planner behavior when evaluated in a closed-loop control setting. We implement these scenarios, visualized in Fig. 3, in the CARLA simulator [14]. Our key hypothesis is that in these scenarios, the ability of the AVP to predict and plan with respect to unobserved vehicles will result in both safe and efficient control compared to existing approaches that do not properly account for unobserved vehicles.

### 4.1   Scenarios

**Two-lane overtake (a):** The ego vehicle attempts to overtake a truck on two-lane freeway and must avoid potential oncoming traffic. An agent that is correctly planning with respect to hypothetical oncoming traffic will make the lane change quickly to avoid getting stuck parallel to the truck with no escape maneuver. An overconfident agent that is oblivious to the possibility of occluded oncoming traffic will attempt the overtake maneuver at a lower speed, opening up the possibility of an unavoidable collision. An underconfident agent will never initiate an overtake.

**Blind summit (b):** The ego vehicle approaches a steep incline occluding the road past the apex of the hill. Because another vehicle may exist at the top of the hill, an expert agent will slow down just enough to avoid an unavoidable collision with a hypothetical occluded vehicle. An overconfident agent oblivious to the possibility of a hidden vehicle may crash depending on their speed, while an underconfident agent will be overly conservative and proceed with a speed far slower than the minimum necessary to avoid a hypothetical collision.

**Intersection (c):** The ego vehicle approaches an intersection with a connecting left turn lane, however the lane is occluded by traffic. An expert agent will slowly proceed through the intersection to avoid a potential collision with an occluded vehicle turning left. An overconfident agent proceeding will cross the intersection faster, but will be unable to avoid a potential collision with an occluded vehicle attempting to make a turn if one exists. An underconfident agent will become frozen prior to entering the intersection due to the inability to see oncoming traffic in the connecting lane.

**Three-agent intersection (d):** In this variation of the intersection, occluded agents may exist to the right and left of the ego vehicle. The behavior of an expert agent is similar to that in the two-agent variant: the ego must proceed through the intersection slowly enough to emergency brake and avoid collisions with other turning vehicles that are initially occluded by traffic.

## 4.2 Training and evaluation procedure

To collect observations to train our learned model, we use hand-coded deterministic policies for all agents, which are designed to generate all plausible modes of realistic driver behavior in the scenario (all modes are balanced equally). At the evaluation stage, we use the same hand-coded deterministic policies for the actors, but use planned trajectory for the ego vehicle with a fixed replanning rate for a fixed horizon $T$. Planned waypoints are turned into explicit low-level actions in the CARLA simulator (steering, throttle and brake) using a P-controller (see the supplement for pseudocode describing the control procedure). We evaluate each method a total of 30 times on each scenario with a balanced split of modes, and report the fraction of episodes where the planner was able to successfully reach the goal (RG) in Tab. 1. Additionally, we report the fraction of episodes where the planner reached the goal in optimal fashion (RG$^*$): for example, an overly conservative planner may successfully clear an intersection (RG), but take much longer in doing so than an experienced human driver (RG$^*$). Joint trajectories x that satisfy RG$^*$ are a strict subset of those that satisfy RG, i.e., RG$^* \rightarrow$ RG. During training and evaluation, we randomize the starting positions of all agents in the scene and randomly drop a fraction of each LiDAR point cloud observation. The CARLA simulator is an additional source of non-determinism (e.g., slight changes in lighting and foliage).

**Single-agent planning baseline:** Though it is possible to plan in multi-agent scenarios using a single-agent behavior model, it will often lead to dangerously overconfident behavior, e.g., entering an intersection while disregarding oncoming traffic. We implement a single-agent (learned) planner baseline using version of our model trained only on ego data (effectively the same model as in [50]).

**Multi-agent fully-observed planning baseline:** In partially observed settings, a multi-agent planner that assumes full observability can lead to dangerous overconfident behavior. For example, the ego vehicle may begin to enter an intersection while oblivious to occluded oncoming traffic; once the ego vehicle begins tracking previously occluded cars, it may be too late to reverse course and avoid a collision. To implement our multi-agent fully-observed baseline, we train our behavioral model on ground-truth fully observed joint trajectories, and only begin predicting a joint trajectory once a non-ego agent has been detected. Similar planners can be created by combining any number of modern trajectory prediction methods (that are flexible to varying numbers of agents) with model-based planning algorithms such as MPC. Our version of the baseline enables planning contingencies over stochastic behavior, however it does not enable planning with respect to undetected agents.

**Multi-agent partially-observed planning baseline:** We compare our approach to a recent game-theoretic planner for partially-observed AV settings proposed by Zhang and Fisac [11] (SOAP), which plans with respect to occluded vehicles by formulating the driving scenario as a two-player zero-sum game of pursuit: the ego vehicle computes an area consisting of all points from which another agent could 'capture' or collide with the ego (the 'danger zone'). The ego vehicle also computes a set of points that a vehicle located just beyond the its field of view could reach in the next $t$ timesteps (the 'forward hidden set'). Having computed both these two sets, the ego vehicle can then safely plan by ensuring the danger zone and forward hidden set never intersect. SOAP generates an expert-level policy similar to AVP, however it requires significant setup per scenario, whereas our method can be easily transferred to new scenarios (assuming expert observations exist).

## 4.3 Results and discussion

In our experiments, AVP clearly outperforms the single-agent and fully-observed multi-agent baselines, both of which exhibit overconfident behavior leading to collisions with occluded vehicles. Since our learned model and planner are robust to noise, the planner either clearly succeeds or clearly fails depending on the scenario configuration: in Tab. 1, we can see that the single-agent and multi-agent baselines will consistently fail when another agent is present (due to each scenario's construction). On the other hand, both multi-agent partially-observed planners (AVP and SOAP [11]) generate expert-level plans that are contingent on the existence of undetected agents.

Although both AVP and SOAP are capable of generating expert-level plans in two-agent partially-observed settings, AVP is a more general solution. While the game-theoretic baseline has no learned

| Method | Two-lane overtake | | | | Blind summit | | | | Intersection (two-agent) | | | | Intersection (three-agent) | | | |
|---|---|---|---|---|---|---|---|---|---|---|---|---|---|---|---|---|
| | RG ↑ | RG* ↑ | C ↓ | Time ↓ | RG ↑ | RG* ↑ | C ↓ | Time ↓ | RG ↑ | RG* ↑ | C ↓ | Time ↓ | RG ↑ | RG* ↑ | C ↓ | Time ↓ |
| Overly conservative reference driver | 30/30 | 0/30 | 0/30 | 26.0s | 30/30 | 0/30 | 0/30 | 25.0s | 30/30 | 0/30 | 0/30 | 23.5s | 30/30 | 0/30 | 0/30 | 28.2s |
| Single-agent (oblivious) [50] | 14/30 | 14/30 | 16/30 | 19.7s | 13/30 | 13/30 | 17/30 | 19.0s | 16/30 | 16/30 | 14/30 | 13.3s | 7/30 | 7/30 | 23/30 | 15.6s |
| Multi-agent (oblivious until too late) [6] | 17/30 | 17/30 | 13/30 | 20.0s | 16/30 | 16/30 | 14/30 | 19.5s | 12/30 | 12/30 | 18/30 | 13.0s | 7/30 | 7/30 | 23/30 | 15.6s |
| Game-theoretic† (designed, 2-agent) [11] | 30/30 | **30**/30 | 0/30 | 24.0s | 30/30 | **30**/30 | 0/30 | 24.0s | 30/30 | **30**/30 | 0/30 | 24.0s | 30/30 | 7/30 | 13/30 | 15.6s |
| Active Visual Planning (ours) | 30/30 | **30**/30 | 0/30 | 24.0s | 30/30 | **30**/30 | 0/30 | 24.0s | 30/30 | **30**/30 | 0/30 | 21.0s | 30/30 | **30**/30 | 0/30 | 25.2s |

**Table 1:** Comparison of our Active Visual Planning (AVP) approach to baselines in a closed-loop control setting. Goal-reaching success rate 'RG' indicates the agent reached the designated goal (higher is better). Near-expert goal-reaching success rate 'RG*' indicates the human-level performance, i.e., the agent reached the designated goal as quickly as the human expert (higher is better). 'C' indicates the number of collisions incurred (lower is better). 'Time' indicates the average time before the goal was reached, *not counting episodes that ended in a collision* (lower is better). † indicates results reported in [11] and via correspondence with the first author.

components (and thus does not require an initial training stage), it requires significant setup for each individual scenario: since the planner requires a discrete state space, the reachable areas of the map must first be discretized into a lanelet map. Additionally, a road centerline for each agent must be specified a priori, since the state of each agent is represented as vertical and lateral positioning relative to a reference trajectory. Computing the forward hidden set during planning requires a precise measure of the ego agent's FOV, which is done by placing (observed) dynamic obstacles in a 3D occupancy grid [51], which must be also generated a priori. While AVP requires training a behavioral model, assuming a dataset of observations exists or can easily be collected, applying AVP to new scenarios is straightforward. AVP predicts and plans continuous positions in Cartesian space, and does not require reference trajectories at test-time based on provided map knowledge. AVP also does not require a geometric representation of FOV to estimate occlusions, and instead learns to recognize occlusions directly from perceptual context, e.g., LiDAR or camera images.

AVP also scales to scenarios with additional agents: the prediction and planning procedures remain the same as the two-agent variant, and the compute required a single forward pass through the neural network scales linearly with the number of agents. In contrast, SOAP exhibits overconfident behavior on the three agent scenario, and thus does not plan safe trajectories (as seen in Tab. 1). Because the SOAP planner uses a game-theoretic solver that is limited to two agents, scaling the existing SOAP planner to three or more agents can only be done by solving multiple two-player games independently between the ego and non-ego agents. This independence assumption leads to overconfident driving policies: for example, the ego may exist in a state with a closed-loop escape policy to avoid car 1, but which results in a collision with car 2 (and vice versa).

**Limitations:** AVP models detection instead of directly modeling existence, which has the key advantage of enabling training on partially-observed data (further discussed in Sec. 3). However, a disadvantage of this approach is reduced interpretability: AVP cannot be probed to visualize the predicted current positions of occluded agents, instead, the model can only express where occluded agents may appear (become disoccluded) in the future. We anticipate future work will further explore this trade-off between interpretability and tractability for planning under perceptual uncertainty. Because AVP uses a learned model, it requires a method of collecting training data in the form of partially-labelled agent coordinates (for agents that are detected by the sensor system), and synchronous perceptual context (such as LiDAR or RGB images). Additionally, the learned trajectory model is object-centric, which (in its current form) does exclude modeling certain phenomena such as objects splitting (e.g., a pedestrian getting out of a vehicle is modeled as a pedestrian appearing out of nowhere). Both of these limitations are shared by most learned trajectory-modeling techniques.

## 5   Conclusion

In this paper we present Active Visual Planning (AVP), a scalable method for contingency planning under multiple forms of partial observability (uncertainty in existence, positions, and intentions) that accommodates unknown dynamics, raw observations, and learning from offline data. Because AVP is learned purely from observations, it does not require expert action labels or domain knowledge (such as state-space discretization or specification of an approximate dynamics model), making it easily generalizable to new environments and scenarios. We demonstrate the effectiveness of AVP on a set of autonomous driving environments inspired by real-world scenarios that require reasoning about the existence of other agents, in which AVP achieves expert-level performance.

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
