# OpenReview forum: "Is Anyone There? Learning a Planner Contingent on Perceptual Uncertainty"
_robot-learning.org/CoRL/2022/Conference — CoRL 2022 Poster_

### Official Review · Reviewer_uLnS · 2022-07-06

**Originality:** Good
**Technical Quality:** Fair
**Clarity Of Presentation:** Excellent
**Impact:** 3

**Recommendation:**

Weak Reject: I recommend rejecting the paper, but will not argue for my recommendation if the majority of other reviewers have a different opinion.

**Summary:**

This paper proposes an imitation learning like method for autonomous driving. The method outputs contingency plans based on maximum likelihood estimation of ego-agent control parameters in a learned probabilistic multi-agent interaction model. The main novelty of the work is to extend the previous PRECOG model (Rhinehart et al., ICCV 2019) by modelling partial observability via detection events. The method is evaluted in three driving scenarios with the CARLA simulator, showing improved performance over baslines using PRECOG or other alternative approaches.

**Issues:**

I would appreciate commenting on/addressing the weaknesses and other comments mentioned above.

**Quality Of The Limitations Section:**

Limitations are not well addressed

**Reviewer Expertise:**

3: The reviewer is fairly confident that the evaluation is correct

**Robotics Focus:**

Highly relevant to robotics but no hardware experiments

**Strengths And Weaknesses:**

** Strengths

- Addresses a highly relevant and interesting problem for autonomous driving: reasoning about future uncertainty and presence of unseen traffic participants, and using this for planning ego-motion.
- Addressing partial observability via modelling detections is interesting, and has the advantage of avoiding need for ground truth positions, which the paper well argues for.
- Paper is overall well written and organized.

** Weaknesses

- The evaluation is rather underwhelming. It focuses on the aggregate performance in the three benchmark tasks, but does not provide any insights into qualitative behavior, or any ablations on the proposed model.

- Especially more effort is needed to evaluate the key novelty of the work, modelling of partial observability. It would be interesting to see how well the model succeeds in predicting future uncertainty, which is key to taking appropriate active sensing actions to potentially reduce that uncertainty. It is not clear what knowledge is represented or how it could be examined/introspected, an empirical demonstration would be welcome. Interpretability is important to have in the target application, but is limited in the proposed method (as is appropriately acknowledge in the paper).

- I also wonder whether the benchmark tasks are too easy, as they are solved perfectly by the proposed method? This gives little clues as to the limits of the proposed method.

- Data association (DA) is an issue in cluttered traffic scenes or with a large number of agents: which measurement originated from which target? DA is assumed solved in the paper, and the detection modelling approach relies on this assumption. It seems unlikely the method can handle complex traffic scenes.

- It seems doubtful the proposed method will scale to more than a handful of agents. Firstly, there is the problem of obtaining training data. The paper is not very clear on how this is done, but Lines 232-233 indicate a substantial amount of manual effort. Second, the paper does not say explicitly, but being based on inference in a graphical model, one could expect scaling to be exponential in the size of the model (but please, correct me if I am mistaken). To be fair, this is a common issue in other methods as well but would be worth commenting on.


** Other comments
- The section on POMDPs and belief space planning is otherwise accurate, but it misses the significant stream of sampling-based Monte Carlo tree search inspired methods, see e.g., POMCP (Silver & Veness, NeurIPS 2010), DESPOT (Somani et al., NeurIPS 2013), and POMCPOW (Sunberg & Kochenderfer, AAAI 2018). These methods have far greater scalability than point-based solvers, and are sometimes applicable in continuous problems. These should be discussed as well.

- Line 102: domain of x_t should be R^{(A+1) x d} to account for ego agent

- How training data is collected is not quite clear (Lines 232-233), this should be properly explained in the paper or supplementary material.

- The paper makes some claims about "closed loop optimality" in the introduction. However, by Section 4.2. it turns out that what is meant is ability to reach human performance (fine, but that's something different). The terminology should be revised, I don't think it is reasonable to claim optimality without a clearly defined reward function.

- What is the relation of this work to methods such as PHD filtering (e.g., Vo & Ma,"The Gaussian mixture probability hypothesis density filter" IEEE Transactions on signal processing)? Is it fair to say the prediction part method approximates a PHD filter with known data association?

**Added after author response**: I suggest the authors also consider the related work Cunningham et al. "MPDM: Multipolicy decision-making in dynamic, uncertain environments for autonomous driving", ICRA 2015, which also models agents with distinctive closed-loop behaviors.

**Summary Of Recommendation:**

This is a work with some novelty (extending PRECOG to partially observable case), but this novel contribution needs a more thorough evaluation. The appropriateness of the evaluation should be checked, and some further insights into what the method has learned or how the knowledge can be examined would be good to have.

**Added after author response**

I thank the authors for their reply. I have also seen the revised paper and supplementary material. This paper has interesting ideas, most notably the approach of modelling detections instead of ground truth positions of other agents. The author response also addresses all the points raised in my "other comments" part well.

However, in my evaluation, the following negative points outweigh the positives:
- Underwhelming evaluation. The evaluation in the main paper simply reports aggregate performance over three scenarios, with perfect scores for the proposed method. This falls short of demonstrating what the limitations of the proposed method are, what are the aspects that are crucial for the proposed method to work (ablation studies), or how it performs with respect to objectives such as maintaining a multi-modal belief over intentions/existence/trajectories of other agents, or actively taking actions to reduce future uncertainty. The revised supplementary material adds qualitative examples, but in my opinion these are key aspects of the proposed method and should be evaluated more thoroughly.
- Lack of scalability. The training data collection requires a manual definition of a control policy for all agents. All combinations of such policies must be present in the training data. Even with 2 distinct behaviours per agent, for n agents the method requires data from 2^n scenarios. This should also be addressed in the limitations, and I agree with Reviewer ec4g that this section can be improved.

Therefore, I am still leaning towards reject.

---

> ### Author Response · Authors · 2022-08-27
> **Response to reviewer uLnS (part 1)**
>
> Thank you for your detailed feedback! We greatly appreciate it. This is a summary of our response:
> - We revised the supplementary material to include additional quantitative results from our benchmark scenarios, as well as additional qualitative results to help visualize the model outputs.
> - We explain how AVP predicts future uncertainty (in agent detection)
> - We explain our assumptions about data association, and how this is similar to prior work in learned trajectory models.
> - We explain how our model scales, and clarify the use of the term inference (used to describe our neural network evaluation, not exact/approximate inference from graphical models).
> - We revised our related work section to include citations to Monte Carla tree search methods.
> - We revised our phrasing of “human optimality” to “expert-level optimality” to avoid confusion with the term “optimality”.
> - We explain the relationship between our work and PHD filtering.
>
> We provide our revised [paper and supplement](https://drive.google.com/drive/u/2/folders/1dwLuCzjXHOzg9-LZd55ADb62FA744FJ8).
>
> The following is our full response:
>
> *“The evaluation is rather underwhelming. It focuses on the aggregate performance in the three benchmark tasks, but does not provide any insights into qualitative behavior, or any ablations on the proposed model.”:*
>
> We revised the supplemental material to include additional quantitative results from our benchmark scenarios (collision rate and time-to-goal). On our supplemental website we provide videos illustrating the policies learned by our model and the baselines (we have revised the supplemental material to include descriptions of the videos). Additionally, we have added some example visuals of generated plans at key decision points to the supplementary material to help visualize the model outputs.
>
> *“It would be interesting to see how well the model succeeds in predicting future uncertainty, which is key to taking appropriate active sensing actions to potentially reduce that uncertainty. It is not clear what knowledge is represented or how it could be examined/introspected, an empirical demonstration would be welcome.”:*
>
> AVP models the future detection probability (and position) of non-ego agents using a Bernoulli random variable. Because AVP models detection using joint positions (and detection) from previous timesteps, AVP enables planning actions to reduce perceptual uncertainty. This is further discussed in section 3 (“modeling partial observability” and “benefits of modeling detection instead of existence”).
> We have revised the supplemental material to include a figure illustrating the output of the model during a transition point in the joint trajectory where uncertainty transitions from being high (the ego is uncertain if the actor exists) to low (the ego is certain the actor exists). This transition in uncertainty is captured by the detection Bernoulli variable - when the ego should be uncertain if the actor exists, the predicted future detection has high entropy (leading to a wide range of samples), and once the ego knows that the actor exists, the predicted future detection has low entropy.
>
> *“Data association (DA) is an issue in cluttered traffic scenes or with a large number of agents: which measurement originated from which target? DA is assumed solved in the paper, and the detection modelling approach relies on this assumption. It seems unlikely the method can handle complex traffic scenes.”*
>
> In our paper, we do not assume the scene is fully observed, however, we do assume that observed agents are correctly tracked. In practice, we add noise to the tracked agent positions to prevent the neural networks from overfitting, however, we do not perturb or randomize the assignment of positions between agents. This assumption is consistent with prior work on learned trajectory models (cited in Figure 1(b)).

---

> ### Author Response · Authors · 2022-08-27
> **Response to reviewer uLnS (part 2)**
>
> (continuing [part 1](https://openreview.net/forum?id=2CSj965d9O4&noteId=fr_lGH-ldKO))
>
>
> *“It seems doubtful the proposed method will scale to more than a handful of agents. Firstly, there is the problem of obtaining training data. The paper is not very clear on how this is done, but Lines 232-233 indicate a substantial amount of manual effort.”*
>
> We have revised the supplemental material to include an additional section on data collection to expand on the discussion in section 4.2, “training and evaluation procedure” (“To collect observations to train … we use hand-coded deterministic policies .. designed to generate all possible modes of realistic driver behavior”).
>
> To clarify, AVP does not depend on the particular method of data collection used, instead, it simply assumes some dataset of partially-observed trajectory data and accompanying perceptual context data (e.g., LiDAR or RGB) exists. In our paper, we aim to evaluate AVP on specific scenarios designed to showcase how AVP can enable contingency planning over partial observability. This requires having training data for these particular scenarios, then running a controlled closed-loop evaluation for the trained model and planner, both training and evaluation thus require hard-coding “AI” policies to mimic human behavior (which does require a substantial amount of manual effort). One could presumably train AVP on a large collection of existing real-world or simulated data containing partially-observed position trajectories and some form of perceptual context (LiDAR, RGB images, etc.), however, evaluating the closed-loop performance of the learned model would require creating a full test environment that matches the training data, which we believe is outside the scope of this paper.
>
> *“Second, the paper does not say explicitly, but being based on inference in a graphical model, one could expect scaling to be exponential in the size of the model (but please, correct me if I am mistaken). To be fair, this is a common issue in other methods as well but would be worth commenting on.”*
>
> Our model is a generative deep neural network (specifically, a recurrent neural network that takes a fixed-length vector input, and has a fixed-length vector output for predicted probability distribution parameters), and is trained using negative log-likelihood with a stochastic gradient descent optimizer (Adam) - further details provided in section 3, “model architecture and training”. To be clear, “inference” in our paper refers to the post-training stage of our DNN, where the network weights are fixed and only used for forward evaluation, not to be confused with traditional exact and approximate inference algorithms for probabilistic graphical models.
>
> Generating joint predictions of future positions using our model (the trained neural network) is constant time and does not depend on the number of agents in a scene because the network and input size are fixed length. During planning, the goal is to generate an optimal trajectory (for the ego/robot) given a cost function; to do so, we run gradient-based optimization (for a fixed number of steps) to set the ego position while sampling the non-ego positions (further details in section 3, “planning with a learned model”). This optimization procedure also does not depend on the number of agents in a scene, and instead only depends on the (constant) time required to evaluate the neural network, and the number of optimization steps to run (a hyperparameter).
>
> *“The section on POMDPs and belief space planning is otherwise accurate, but it misses the significant stream of sampling-based Monte Carlo tree search inspired methods, see e.g., POMCP (Silver & Veness, NeurIPS 2010), DESPOT (Somani et al., NeurIPS 2013), and POMCPOW (Sunberg & Kochenderfer, AAAI 2018). These methods have far greater scalability than point-based solvers, and are sometimes applicable in continuous problems. These should be discussed as well.”*
>
> We’ve revised the related work on POMDPs and belief space planning to include the recommended references.
>
> *“The paper makes some claims about "closed loop optimality" in the introduction. However, by Section 4.2. it turns out that what is meant is ability to reach human performance (fine, but that's something different). The terminology should be revised, I don't think it is reasonable to claim optimality without a clearly defined reward function.”*
>
> We’ve revised our statements about optimality to be clear that “optimal” means “expert-level performance”.

---

> ### Author Response · Authors · 2022-08-27
> **Response to reviewer uLnS (part 3)**
>
> (continuing [part 2](https://openreview.net/forum?id=2CSj965d9O4&noteId=5KJsmHQUfV0))
>
> *“What is the relation of this work to methods such as PHD filtering (e.g., Vo & Ma,"The Gaussian mixture probability hypothesis density filter" IEEE Transactions on signal processing)? Is it fair to say the prediction part method approximates a PHD filter with known data association?”*
>
> Our method (AVP) is less structured than PHD filtering: rather than using a predict and update step explicitly in a belief space, we learn a mapping from a history of observations to the next observation. This is related to our choice to model agent detection rather than agent existence, which we discuss in more detail in a subheading in Section 3. Full PHD filtering, with an explicit estimate of the underlying true state of the environment, would thus be an improvement to our method, which we will add as suggested future work. Though we recognize PHD filtering would improve our work, we also wish to stress the focus of this work was not the filtering (or lack thereof) but that we can reason about future observations of other agents given high dimensional scene context.
>
> Turning specifically to the question of data association in the prediction step, it is true that we assume we can associate future vehicle positions with the correct past trajectory and sampled noise variable z. This is a consequence of us staying in a purely data-driven regime and not explicitly stating an environmental belief space model, which might include a measurement likelihood that accounts for uncertainty in vehicle tracking. This sort of accurate measurement likelihood in an autonomous vehicle setting would involve a model of the object tracking performance in the perception component of an AV system, which is outside of the scope of this work.

---

### Official Review · Reviewer_ec4g · 2022-07-22

**Originality:** Good
**Technical Quality:** Good
**Clarity Of Presentation:** Very Good
**Impact:** 3

**Recommendation:**

Weak Reject: I recommend rejecting the paper, but will not argue for my recommendation if the majority of other reviewers have a different opinion.

**Summary:**

This paper develops a planner for autonomous driving in a multiagent environment. In particular, the paper contributes a method for addressing an imperfect perception system (e.g., sometimes doesn't detect other agents). A learned model of detection of other agents is fed into a planner that aims to compute collision-free paths that reach the goal quickly. Experiments in CARLA simulations of 4 traffic scenarios suggest the proposed method performs better than three prior approaches.

**Issues:**

- Table 1 should be re-generated in a more sensible manner. RG and RG* are super vague concepts that throw away a lot of information here. The important metrics here seem to be reaching the goal quickly and not colliding despite the imperfect knowledge. I suggest reporting how each algorithm does on these metrics (and/or other metrics), and then incorporate a human expert either as a normalization factor (e.g., time to goal relative to human expert) or as another "algorithm" baseline row. Also, since every algorithm gets 15, 30, or 60 out of 60, it suggests there are a few discrete jumps in the difficulty of the scenarios --> this should be elaborated on because it seems unlikely to be a coincidence.
- Additional experiments should be done (e.g., suggestions above) to provide a deeper level of analysis/insights into how uncertainty and the planner interact.
- Block diagram should be improved per comments above.
- The results section claims AVP scales to scenarios with additional agents, but the experiments show a jump from 2 to 3 agents. This is not a strong demonstration of scalability.
- Modify overclaiming/inaccurate statement in the conclusion "Because AVP is learned purely from observations, it does not require expert action labels or domain knowledge (such as state-space discretization or specification of an approximate dynamics model), making it easily generalizable to new environments and scenarios", since "we use hand-coded deterministic policies for all agents, which are designed to generate all plausible modes of realistic driver behavior in the scenario."
- It may be worth adding a reference to the work of Charles Richter (papers circa 2015) that deal with planning conservatively around blind corners (a related form of perceptual uncertainty)?
- The limitations section is not being used as intended. The first sentence describes an advantage of the approach. The third sentence describes future work. Surely there are more limitations than a lack of interpretability.

**Quality Of The Limitations Section:**

Limitations are not well addressed

**Reviewer Expertise:**

4: The reviewer is confident but not absolutely certain that the evaluation is correct

**Robotics Focus:**

Highly relevant to robotics but no hardware experiments

**Strengths And Weaknesses:**

Strengths:
- Important and difficult problem, relevant to this community, well-motivated
- Modeling partial observability as a Bernoulli variable of whether an agent will be detected makes sense
- Incorporating this model of perceptual uncertainty into the planner makes sense

Weaknesses:
- Results section has a lot of words but not much substance. Only a high-level comparison is given on 4 relatively clear scenarios. Since perceptual uncertainty is such an important factor, this section would have been much stronger with more experimental analysis of the detection model, how much uncertainty the detector/planner can handle, how realistic these uncertainties are (maybe using real driving data), etc. Issue with Table 1 described further below.
- Block diagram in Figure 2 should be re-structured. What is the overall output? Where does the planner connect? I don't understand this diagram.

**Summary Of Recommendation:**

Overall this is a cool problem and the approach seems promising, but the experimental results do not provide sufficiently deep insight into what's really going on here.

---

> ### Author Response · Authors · 2022-08-27
> **Response to reviewer ec4g (part 1)**
>
> Thank you for your detailed feedback! We greatly appreciate it. This is a summary of our response:
> - We explain how our existing metrics are normalized by human experts.
> - We explain the discrete jumps in performance of the planners.
> - We explain that our results on three agent scenarios is an improvement over prior work.
> - We explain how hand-coded policies are used for data collection and evaluation, and that AVP does not inherently depend on the existence of hand-coded policies.
> - We revised the related work to include the suggested references.
> - We revised the limitations section to include additional limitations.
>
> We provide our revised [paper and supplement](https://drive.google.com/drive/u/2/folders/1dwLuCzjXHOzg9-LZd55ADb62FA744FJ8).
>
> The following is our full response:
>
> *“Block diagram in Figure 2 should be re-structured. What is the overall output? Where does the planner connect? I don't understand this diagram.”*
>
> The overall output of the model is the “predicted distribution of future states X for each agent and detections D for each non-ego agent” (from the figure caption). This is visualized by the “trajectory decoder” in the figure: the red circle represents the predicted future non-ego agent position (for which AVP must model its detection), and the blue circle represents the predicted future ego position. This sequence (of predicted future positions) is autoregressively generated up to a maximum trajectory length (horizon), and the full joint trajectory is represented by the “predicted future trajectories” column on the right. The planner optimizes the free parameters z for the ego vehicle (further discussed in section 3, “Planning with a learned model”) - both the planner and the robot z variable in the figure is highlighted in orange to visualize this connection.
>
> *“I suggest reporting how each algorithm does on these metrics (and/or other metrics), and then incorporate a human expert either as a normalization factor (e.g., time to goal relative to human expert)”*
>
> To clarify, our reach-goal RG* metric is already normalized by a human expert: RG* indicates whether the planner reached the goal in an optimal fashion consistent with a human driver (as described in section 4.2).
>
> *“since every algorithm gets 15, 30, or 60 out of 60, it suggests there are a few discrete jumps in the difficulty of the scenarios --> this should be elaborated on because it seems unlikely to be a coincidence”*
>
> To clarify, the reason we observe even fractions in the performance of the planners on each scenario is because there is an even split in the “modes” (variations) of each scenario (described in section 4.2, “all modes are balanced equally”). As mentioned in section 4.3, “the single-agent and multi-agent baselines will consistently fail when another agent is present (due to each scenario’s construction)”, which leads to the “discrete jumps” in Table 1.
>
> *“The results section claims AVP scales to scenarios with additional agents, but the experiments show a jump from 2 to 3 agents. This is not a strong demonstration of scalability.”*
>
> Our experiments on scenarios with 3 agents is a demonstration in scalability relative to existing prior work, which only demonstrated contingency planning using two agents.
>
> *"Modify overclaiming/inaccurate statement in the conclusion Because AVP is learned purely from observations, it does not require expert action labels or domain knowledge (such as state-space discretization or specification of an approximate dynamics model), making it easily generalizable to new environments and scenarios", since "we use hand-coded deterministic policies for all agents, which are designed to generate all plausible modes of realistic driver behavior in the scenario."*
>
> To clarify, we only use hand-coded policies to generate training data and control the non-controllable/non-ego cars in the evaluation - there is nothing inherent to AVP that requires hand-coded policies (the second statement does not invalidate the first statement). We use CARLA for our training data and evaluation because it allows us to design scenarios that specifically illustrate the key contribution of our approach (planning under perceptual uncertainty). This is consistent with prior work in learned trajectory planners that use simulators such as CARLA to evaluate their method (they either use hand-coded policies to control the non-ego agents in the scene, or to create training data for their learned model, or both).

---

> ### Author Response · Authors · 2022-08-27
> **Response to reviewer ec4g (part 2)**
>
> (continuing [part 1](https://openreview.net/forum?id=2CSj965d9O4&noteId=v0MEFjjyEQ9))
>
> *“It may be worth adding a reference to the work of Charles Richter (papers circa 2015) that deal with planning conservatively around blind corners (a related form of perceptual uncertainty)?”*
>
> We have revised the related work to include references to Richter et al. 2014 (“High-speed autonomous navigation of unknown environments using learned probabilities of collision”) Richter et al. 2015 (“Bayesian Learning for Safe High-Speed Navigation in Unknown Environments”) in our discussion of methods for “Occlusion-aware prediction and planning”.
>
> *“The limitations section is not being used as intended … surely there are more limitations than a lack of interpretability”*
>
> We have revised the limitations section to include additional discussion of limitations, which include the limitations of using a learned model (it requires a method to collect training data), as well as the object-centric focus of our modeling approach (both are limitations shared by most learned trajectory-modeling techniques).

---

### Official Review · Reviewer_56zH · 2022-07-30

**Originality:** Good
**Technical Quality:** Very Good
**Clarity Of Presentation:** Very Good
**Impact:** 4

**Recommendation:**

Weak Accept: I recommend accepting the paper, but will not argue for my recommendation if the majority of other reviewers have a different opinion.

**Summary:**

The authors propose Active Visual Planning (AVP), a data-driven approach for automated driving that uses context derived from perceptual observations (specifically, lidar scans) along with past vehicle waypoints to predict the distributions of future states for both an ego vehicle and non-ego vehicles, enabling contingency planning which is capable of addressing the partial observability that arises in common driving scenarios. Learning occurs offline, using a dataset of recorded observations, and hand-coded deterministic policies reflective of human driving behaviors. Per the study conducted using the CARLA simulator, AVP permits the scenarios of interest to be handled with increased safety, relative to single and multi-agent models that plan only according to the information that is observable. The authors claim that AVP is the first fully-learned planner that supports active contingency planning over perceptual uncertainty.

**Issues:**

- The authors should provide more evidence to support the claim “our method can be easily transferred to new scenarios”. Perhaps they can do so by providing clarification on the extent to which the driving scenarios encountered in the training process differed (or did not differ) from the scenarios encountered at test-time. For example, what was different about the ego and non-ego trajectories, environments, etc., encountered in training vs. testing.

- Along these lines, the statement “further details can be found in the supplemental material” in the section on Model Architecture and Training gives the reader the impression that more details on the training procedure will be found in the supplement, but there are no additional details provided.

- The term “visual context” is a little confusing, since vision is not one of the sensing modalities being used in this work. Perhaps “perceptual context” could be used instead for greater generality, or “scan context” for greater specificity. If visual context is a term of art commonly used to describe this, feel free to ignore my suggestion.


**Quality Of The Limitations Section:**

Limitations are addressed clearly

**Reviewer Expertise:**

4: The reviewer is confident but not absolutely certain that the evaluation is correct

**Robotics Focus:**

Highly relevant to robotics but no hardware experiments

**Strengths And Weaknesses:**

+ The paper addresses a compelling, well-motivated topic relevant to robot learning, proposing a novel method for predicting and acting upon the existence, positions, and intentions of other vehicles on the road under partial observability, to safely complete key automated driving tasks.

+ The paper is well-organized and clearly written, the code for AVP is made available, and the supplemental details provided on the accompanying website are helpful.

+ AVP is compared with three other suitably chosen methods, which include simpler, less conservative frameworks (single and multi-agent models that plan only according to the information that is observable) and a game-theoretic framework that considers all of the possible outcomes of the driving scenarios of interest.

+ The driving scenarios selected for the experimental study are well-motivated scenarios that arise in everyday driving, which also present safety hazards for human drivers. AVP is shown to address these driving scenarios with superior rates of success relative to the selected baselines.

- It would be helpful to provide more detailed, quantitative results characterizing the plans generating by the four competing methods, as well as to show, qualitatively, what the plans look like that were produced by the three baselines selected.

- The paper would be strengthened by providing clarification on the extent to which the driving scenarios encountered in the training process differed (or did not differ) from the scenarios encountered at test-time. This would lend support to the claim made in the paper that AVP is flexible and can be easily transferred to new scenarios. This is especially important because of the additional claims made that one of the baselines (SOAP) requires significant setup for each individual scenario.

- The paper would also be strengthened by speculating on the pathway by which the planners learned for specific driving scenarios could be integrated into a single unified framework for use in real automated driving.


**Summary Of Recommendation:**

In AVP, the authors have contributed a novel and well-motivated data-driven framework for prediction and planning that is of potential benefit to complex and hazardous automated driving scenarios, offering improved performance with respect to successful task completion. The paper would be strengthened by providing more implementation details, a more detailed exposition of the experimental results, and evidence for the flexibility/transferability of AVP that is claimed by the authors.

Post-Rebuttal Comments: I appreciate the revisions and clarification provided by the authors, and I remain in favor of acceptance. Taking the authors' responses and the CoRL review criteria into account, I have decided to keep my rating of weak accept.

---

> ### Author Response · Authors · 2022-08-27
> **Response to reviewer 56zH**
>
> Thank you for your detailed feedback! We greatly appreciate it. This is a summary of our response:
> - We revised the supplemental material to include additional quantitative and qualitative results characterizing the plans generated by each method on the scenarios.
> - We revised the main paper (section 4.2) to include additional information about how the experiments were randomized.
> - We revised the supplemental material to include additional details on the model training procedure.
> - We revised the main paper to use the term “perceptual context” instead of “visual context” to better emphasize the generality of the context used in AVP.
> - We explain why SOAP has a larger burden on additional setup for new scenarios compared to AVP.
> - We explain how AVP may be integrated into real-world systems.
>
> We provide our revised [paper and supplement](https://drive.google.com/drive/u/2/folders/1dwLuCzjXHOzg9-LZd55ADb62FA744FJ8).
>
> The following is our full response:
>
> *“It would be helpful to provide more detailed, quantitative results characterizing the plans generating by the four competing methods, as well as to show, qualitatively, what the plans look like that were produced by the three baselines selected.”*
>
> We have added some example visuals of generated plans at key decision points to the supplementary material to help visualize the model outputs (note that closed-loop execution of these plans results in the same results shown in the existing videos).
>
> *“The paper would be strengthened by providing clarification on the extent to which the driving scenarios encountered in the training process differed (or did not differ) from the scenarios encountered at test-time.”*
>
> We’ve revised section 4.2 to include information about how the scenarios were randomized (the global random seed, positions of all agents, and LiDAR observations are randomized, in addition to extra non-determinism induced by the CARLA simulator / Unreal Engine). The foundation of our approach in an imitative model, which expects that the scenarios seen at test time are similar to those seen during training time (i.e., to perform optimally in a scenario, AVP must have been trained on a dataset that observed an optimal human driver in a similar scenario).
>
> *“This is especially important because of the additional claims made that one of the baselines (SOAP) requires significant setup for each individual scenario.”*
>
> To clarify, the burden on additional setup for new scenarios is much higher for SOAP than for AVP: AVP only requires collecting new behavioral data for the new scenario (partially-observed trajectories and LiDAR/visual context snapshots), whereas SOAP requires mapping, labeling road geometry, and a host of other labor-intensive steps (further described in section 4.3).
>
> *“The paper would also be strengthened by speculating on the pathway by which the planners learned for specific driving scenarios could be integrated into a single unified framework for use in real automated driving.”*
>
> We are happy to elaborate on potential integration of methods such as AVP into real-world systems. To the best of our knowledge, there is no published work on learned contingency planners that handle partial observability - AVP demonstrates one way to do this tractably (by modeling future detection). This may be a good modeling decision for real-world systems that use learned trajectory planners that are trained on real-world data, since most real-world data is naturally labeled under the assumption of partial-observability. Any time you have object detection labels that are used in a detect-and-plan system, and instead of doing heuristics for partial observability, you could incorporate a learned planner such as AVP to learn how to deal with partial observability instead.
>
> *“Model Architecture and Training gives the reader the impression that more details on the training procedure will be found in the supplement, but there are no additional details provided.”*
>
> We revised the supplement to include additional details on the model training procedure (extending our existing description of the training procedure in section 3, “Model architecture and training”).
>
> *“The term “visual context” is a little confusing, since vision is not one of the sensing modalities being used in this work. Perhaps “perceptual context” could be used instead for greater generality, or “scan context” for greater specificity.”*
>
> We agree that “perceptual context” better emphasizes the generality of the context used in AVP (i.e., that it doesn’t depend on LiDAR data existing in the observations), and have revised the paper to replace “visual context” with “perceptual context”.

---

### Official Review · Reviewer_D5KJ · 2022-07-30

**Originality:** Fair
**Technical Quality:** Very Good
**Clarity Of Presentation:** Fair
**Impact:** 3

**Recommendation:**

Weak Accept: I recommend accepting the paper, but will not argue for my recommendation if the majority of other reviewers have a different opinion.

**Summary:**

This paper presents an approach to safety-aware trajectory planning in the presence of both seen and unseen agents, with a focus on simulated self-driving scenarios. Building upon earlier work, in which an RNN is used to estimate trajectories. In the (previous) approach, future trajectories of the human agents are sampled from a normal distribution trained from data collected offline and the trajectories of the agent are chosen via optimization with respect to those sampled trajectories. In this work, the authors train the system in such a way that the other "human" agents may not always be visible and so predictions about the true state of the system must incorporate that other agents may be in the scene yet not currently observed. The proposed agent is capable of understanding the impact unseen agents may have on their planning in a handful of common driving scenarios, improving performance with respect to baseline approaches that do not consider this term.


**Issues:**

The central issue is the clarity of the paper's description of the overall approach, mainly the omission of the planning pseudocode. As mentioned above, though what is included in the paper is clear, it was difficult to discern how everything came together during planning; the pseudocode helped my understanding of this, though there are perhaps other ways to communicate these ideas as well.

There are other small questions above that, if included, would also help clarity.

**Quality Of The Limitations Section:**

Additional details required

**Reviewer Expertise:**

3: The reviewer is fairly confident that the evaluation is correct

**Robotics Focus:**

Relevant but unlikely to deploy to hardware in near future

**Strengths And Weaknesses:**

Overall, this paper has an interesting (if somewhat small) idea about how to incorporate unseen agents in decision-making. The main contribution seems mostly to be the change in the form of the data and the training process, which now takes into consideration that not all vehicles will be observed at all times and so the unobserved state should not be directly penalized; instead, the latent state used for forward simulation of trajectories during planning consumes the partial state. While this approach is novel, the contribution seems fairly small. The application domain, however, is fairly important and understudied, offsetting this.

The prose in the paper is very clear. However, the paper was hard to follow in places, due primarily to a notable omission: the pseudocode describing the planning procedure exists only in the supplementary material, yet was important for understanding the method. Without it, the paper is not particularly self-contained and so the authors must spend more time describing what this approach looks like in the main text or move the pseudocode into the text; while the equations listed in the text are enough to define the optimization objective, it is a big leap from following along with those equations. Reading through the pseudocode was essential for my understanding of the proposed approach.
Adding in the pseudocode would greatly help understanding.

Smaller questions and comments for clarity.
- Watching the overtake videos in the supplemental material, it appears that (to at least a certain extent), the ability of the agent to succeed during the overtake boils down partially to luck. There does not appear to be sufficient time to successfully overtake if the oncoming car were to arrive sooner. Have the authors experimented with the timing of the arrival of the oncoming car and do they expect this would change the results? (There is a balance here between seeking opportunity and behaving conservatively, and I still believe the results to show relative improvement to the other results. My question touches upon the agent's desire to prioritize safety: is this a tunable parameter, or would it require changing the training data?)
- The experiments seem to be broken into "chunks" of 15, since every result for all planners succeed in 15, 30, or 60 of the 60 trials. Relatedly, is each trial within each experiment identical or are the trials randomized somehow? The authors should include clarity about how these trials are run in the results.


**Summary Of Recommendation:**

While this approach is novel, the contribution seems fairly small. The application domain, however, is fairly important and understudied. The central issue with the paper as it exists right now is in its clarity, and the overall approach was difficult to fully understand without multiple full read throughs of the paper and reading the supplemental material.

---

> ### Author Response · Authors · 2022-08-27
> **Response to reviewer D5KJ**
>
> Thank you for your detailed feedback! We greatly appreciate it. This is a summary of our response:
> - We revised the paper to include the planning pseudocode / algorithm block in the main text
> - We explain how our experiments consider variable timing of the oncoming car
> - We explain more about the nature of the safety parameterization of the cost function
> - We explain how our experiments are randomized
>
> We provide our [revised paper and supplement](https://drive.google.com/drive/u/2/folders/1dwLuCzjXHOzg9-LZd55ADb62FA744FJ8).
>
> The following is our full response:
>
> Regarding your concern about the lack of pseudocode in the main text causing issues with clarity: we have revised the main manuscript to incorporate the planning pseudocode. Please see the revised section 3 (“Planning with a learned model”), which has been modified to refer directly to embedded pseudocode/algorithm block.
>
> *“Have the authors experimented with the timing of the arrival of the oncoming car and do they expect this would change the results?”*
>
> In our experimental setup, we randomized the starting positions of all vehicles (both during training and evaluation). Assuming enough data exists to train the imitative model, our method should behave just as safely (correctly choosing the optimal plan, i.e., mimicking an expert human driver) given further randomizations in vehicle positions or other environment settings.
>
> *“My question touches upon the agent's desire to prioritize safety: is this a tunable parameter, or would it require changing the training data?)”*
>
> During planning, the cost function can be tuned to prioritize (or deprioritize) safety. As described in section 3 (above equation (5)), we apply a near-collision penalty as an additional constraint (on top of the prior and goal-reaching terms). To prioritize safety, one can simply reweight the terms in this equation before running the optimizer during planning (e.g., by up-weighting the near-collision penalty). Similarly, one can add additional constraints, such as speed or jerk thresholds (to improve ride comfort), which will push the model towards preferring certain driving styles seen in the training dataset.
>
> *“Is each trial within each experiment identical or are the trials randomized somehow?”*
>
> As previously mentioned, the starting positions of all agents in the scene are randomized. We randomly sample a subset of the LiDAR ray casts (during training and evaluation) to prevent overfitting to sensor data. We also sample a global random seed for each run (the CARLA simulator is non-deterministic and adds additional randomness). Combined, this allows us to account for seed randomization for each scenario configuration. We have revised section 4.2 to include this information.

---

### Meta-Review · Area_Chair_aEUk · 2022-08-09

**Recommendation:** Accept (Poster)
**Confidence:** 4

**Metareview:**

The paper addresses the problem of trajectory planning in scenarios where some of the agents may be partially observed. The central idea in this paper is learning of a latent representation that via forward simulation can explain missing observations from agents that may not be observed at all times. The work builds incrementally on a closely related prior work and extends it to the partial-observability setting.

The authors present results on simulated self-driving scenarios and provide comparison with a benchmark game theoretic approach. The primary weakness of the paper lies in the results section. Reviewers expressed their concern that the results could have better explored the nature and extent of uncertainty that the proposed method can handle. Further, a more detailed technical exposition for the planning component was required for the paper to be self standing.

During the rebuttal phase, the authors provided clarification on the experimental setup and relationship with other alternative approaches. The reviewers appreciate the inclusion of the pseudo-code in the manuscript and the additional experimental details. However, it is still felt the experimental evaluation remains a weakness of the paper. In particular, the results are very good on the specific scenarios provided. It is less clear how the approach would generalise to other diverse scenarios. Further, authors are requested to additionally consider providing ablation analysis to better situate the contribution of individual components of the proposed approach.